 

# Basis of specificity for a conserved and promiscuous chromatin remodeling protein

Drake A Donovan[1], Johnathan G Crandall[1], Vi N Truong[1], Abigail L Vaaler[1], Thomas B Bailey[1], Devin Dinwiddie[1], Orion GB Banks[1], Laura E McKnight[1]*, Jeffrey N McKnight[1,2]

[1]Institute of Molecular Biology, University of Oregon, Eugene, United States; [2]Phil and Penny Knight Campus for Accelerating Scientific Impact, University of Oregon, Eugene, United States

**Abstract** Eukaryotic genomes are organized dynamically through the repositioning of nucleosomes. Isw2 is an enzyme that has been previously defined as a genome-wide, nonspecific nucleosome spacing factor. Here, we show that Isw2 instead acts as an obligately targeted nucleosome remodeler in vivo through physical interactions with sequence-specific factors. We demonstrate that Isw2-recruiting factors use small and previously uncharacterized epitopes, which direct Isw2 activity through highly conserved acidic residues in the Isw2 accessory protein Itc1. This interaction orients Isw2 on target nucleosomes, allowing for precise nucleosome positioning at targeted loci. Finally, we show that these critical acidic residues have been lost in the *Drosophila* lineage, potentially explaining the inconsistently characterized function of Isw2-like proteins. Altogether, these data suggest an 'interacting barrier model,' where Isw2 interacts with a sequence-specific factor to accurately and reproducibly position a single, targeted nucleosome to define the precise border of phased chromatin arrays.

**\*For correspondence:**
lthom009@gmail.com

**Competing interests:** The authors declare that no competing interests exist.

## Introduction

Chromatin consists of the nucleic acids and proteins that make up the functional genome of all eukaryotic organisms. The most basic regulatory and structural unit of chromatin is the nucleosome. Each nucleosome is defined as an octamer of histone proteins, which is wrapped by approximately 147 base pairs of genomic DNA (*Luger et al., 1997*; *Kornberg, 1974*). The specific positioning of nucleosomes on the underlying DNA can have significant effects on downstream processes, such as promoter accessibility and molecular recruitment, which ultimately serve to alter gene expression (*Lai and Pugh, 2017*). Despite decades of research, the mechanisms leading to precise nucleosome locations in cells are still being defined.

Nucleosome positioning is dynamically established by a group of enzymes known as ATP-dependent chromatin remodeling proteins (ChRPs) (*Zhou et al., 2016*). Extensive biochemical and structural characterization has been performed on this group of proteins from various families (*Clapier et al., 2017*). The *ch*romodomain-*h*elicase-*D*NA binding (CHD) and *i*mitation *sw*itch (ISWI) families of ChRPs have been characterized as nonspecific nucleosome sliding and spacing factors in vitro (*Stockdale et al., 2006*; *Hauk et al., 2010*; *McKnight et al., 2011*; *Kagalwala et al., 2004*; *Lusser et al., 2005*; *Tsukiyama et al., 1999*; *Pointner et al., 2012*). In yeast, flies, and mammals, ChRPs generate evenly spaced nucleosome arrays at transcription start sites and organize genomic chromatin at other defined boundaries (*Pointner et al., 2012*; *Lee et al., 2007*; *Mavrich et al., 2008a*; *Valouev et al., 2011*; *Krietenstein et al., 2016*; *Wiechens et al., 2016*; *Baldi et al., 2018*; *Gkikopoulos et al., 2011*; *Zhang et al., 2011*). However, relatively little is known about the in vivo

**eLife digest** DNA encodes the genetic instructions for life in a long, flexible molecular chain that is packaged up neatly to fit inside cells. Short sections of DNA are wound around proteins to form bundles called nucleosomes, and then spun into chromatin fibres, a more compact form of DNA. While nucleosomes are a fundamental part of this space-saving packaging process, they also play a key regulatory role in gene expression, which is where genes are decoded into working proteins.

Placing nucleosomes at regular intervals along DNA invariably controls which parts of the DNA – and which genes – the cell's machinery can access and 'read' to make proteins. But the nucleosomes' positions are not fixed, and gene expression is a dynamic process. The cell often uncoils and repackages its DNA while molecular motors called chromatin remodelling proteins move nucleosomes up and down the DNA, exposing some genes and obstructing others.

One group of chromatin remodelling proteins are called Imitation Switch (ISWI) complexes. It has long been thought that these complexes position nucleosomes with little regard to the underlying DNA sequence or the genes encoded, that is to say in a non-specific way. However, this theory has not been thoroughly tested. It is possible that ISWI complexes actually place nucleosomes on certain parts of DNA at particular times in an organism's development, or in response to other environmental factors. Except how such precision is achieved remains unknown.

To test this alternative theory of nucleosome positioning, Donovan et al. studied ISWI proteins and nucleosomes in common baker's yeast. This involved systematically removing sections of ISWI proteins to see whether the complexes could still position nucleosomes, and which parts of the proteins where essential for the job. By doing so, Donovan et al. identified multiple 'targeting' proteins that bind to ISWI proteins and deliver the complexes to specific target sequences of DNA. From there, the complex remodels the nucleosome, positioning it at a specific distance from its landing site on DNA, as further experiments showed.

This research provides a new model for explaining how nucleosomes are positioned to package DNA and control gene expression. Donovan et al. have identified a new mechanism of interaction between nucleosomes and chromatin remodelling proteins of the ISWI variety. It is possible that more interactions of this kind will be discovered with further research.

biological regulation of these spacing factors, and it is not understood how they can accurately and reproducibly position nucleosomes throughout the genome in different cellular contexts.

A widely accepted model is that ChRPs pack nucleosome arrays against a noninteracting barrier, such as an unrelated DNA binding protein or another nucleosome (*Krietenstein et al., 2016*; *Zhang et al., 2011*; *Mavrich et al., 2008b*). In this way, general regulatory factors (GRFs) could establish chromatin landscapes with differing nucleosome arrays in response to changes in the cellular environment. In support of this model, nucleosome arrays near GRFs and other DNA binding elements appear to be phased relative to the binding motifs of the sequence-specific DNA binding factors in cells and in biochemically reconstituted cell-free systems (*Krietenstein et al., 2016*; *Baldi et al., 2018*; *Yan et al., 2018*). This model suggests that boundaries of nucleosome arrays are determined by the binding of barrier factors. Implicit in this barrier model are the assumptions that ChRPs act as nonspecific nucleosome spacing machines throughout the genome and that specific ChRP and GRF interactions are not required to establish nucleosome positions. While this model provides a good explanation for how phased nucleosome arrays can be established throughout the genome by a combination of DNA binding factors and nonspecific chromatin remodeling factors, the fundamental assumptions of the barrier model have not been thoroughly tested.

It has been shown through genetic and recent biochemical experiments that members of the ISWI family of ChRPs functionally interact with transcription factors in vivo (*Krietenstein et al., 2016*; *Gelbart et al., 2005*; *Goldmark et al., 2000*; *Fazzio et al., 2001*; *Yadon et al., 2013*). One of the most well-defined interacting partners of ISWI proteins is the meiotic repressor *u*nscheduled *me*iotic gene *e*xpression (Ume6), which is found in yeasts. It has been previously demonstrated that Ume6 and Isw2, an ISWI-containing ChRP complex in *Saccharomyces cerevisiae* (homologous to the ATP-dependent *c*hromatin *a*ssembly *f*actor [ACF] complex in humans and flies), share genetic targets of

repression and likely interact physically (*Goldmark et al., 2000*). While interactions with sequence-specific DNA binding proteins can potentially determine precise nucleosome targeting and final nucleosome positions (*Donovan et al., 2019*; *Bowman and McKnight, 2017*; *McKnight et al., 2016*), the mechanisms through which physical interactions between Isw2 and any genomic recruitment factor like Ume6 influence nucleosome positioning activity in cells have not been defined. For example, it is not known how these physical interactions occur or what role they play in the biochemical outcomes of chromatin remodeling reactions and the resulting downstream biological outputs.

In this work, we have successfully identified the mechanism of interaction between Isw2 and Ume6 in *S. cerevisiae*. By taking a protein dissection approach combined with genome-wide nucleosome profiling, we have identified a previously uncharacterized helical domain in Ume6 that allows for Isw2 binding, specific genomic recruitment, and precise nucleosome positioning outcomes. We further demonstrate that conserved attributes of this helical domain are observed in the cell cycle regulator Swi6, which we have identified as a new Isw2-recruitment adapter protein that allows for specific nucleosome positioning at Mbp1/Swi6 (MBF) and Swi4/Swi6 (SBF) targets. We have also determined that the transcription factor-interacting interface of Isw2/ACF-like remodeling complexes contains a few key and highly conserved residues within the WAC (*WTSF/Acf1/cbp146*) domain. Finally, we show that these residues, which are essential for directional, sequence-specific remodeling, were lost in the evolution of the *Drosophila* lineage, where extensive biochemical, genetic, and genomic characterization has been performed on the *ITC1* ortholog ACF.

## Results

### Isw2 activity in cells is inconsistent with known biochemistry and the barrier model for nucleosome packing

We wished to understand how the conserved Isw2 protein complex in yeast behaves genome-wide and at specific promoter nucleosomes at target sites. Yeast Isw2 has been characterized extensively in biochemical assays, which all suggest that it has nonspecific DNA binding, ATP hydrolysis, nucleosome sliding, mononucleosome centering, and nucleosome spacing activities (*Stockdale et al., 2006*; *Kagalwala et al., 2004*; *Lusser et al., 2005*; *Tsukiyama et al., 1999*; *Dang and Bartholomew, 2007*; *Dang et al., 2006*; *Hota et al., 2013*; *Kassabov et al., 2002*; *Zofall et al., 2004*; *Zofall et al., 2006*). These nonspecific nucleosome mobilizing activities suggest that the Isw2 protein should be able to organize nucleosome arrays against a barrier across the genome in yeast cells since (1) it is estimated that there are enough Isw2 molecules for every 10–20 nucleosomes in the genome (*Gelbart et al., 2005*), (2) *Drosophila melanogaster* ACF, an Isw2 ortholog, can organize nucleosomes into evenly spaced arrays (*Baldi et al., 2018*), and (3) other nonspecific and related nucleosome spacing factors can globally space nucleosomes across the genome in yeast and other organisms (*Pointner et al., 2012*; *Wiechens et al., 2016*; *Gkikopoulos et al., 2011*; *Zhang et al., 2011*). To first determine how Isw2 positions nucleosomes in *S. cerevisiae*, we examined nucleosome positioning activity in an *isw1/chd1* deletion background to remove known and potentially overlapping global spacing factors and highlight 'isolated positioning activity' by Isw2. When examining the positioning of nucleosomes with and without Isw2 at all yeast pre-initiation complex sites (PICs), it is evident that Isw2 activity is specialized at only a subset of target sites (*Figure 1A*). As seen previously (*Gkikopoulos et al., 2011*; *Ocampo et al., 2016*), no global nucleosome spacing or organizing activity is detected by Isw2 alone (*Figure 1—figure supplement 1*). Close inspection of Isw2-targeted PICs suggests that Isw2 can only organize a single PIC-proximal nucleosome, while subsequent nucleosomes become more poorly phased as the distance from the initially positioned nucleosome increases (*Figure 1A*, *Figure 1—figure supplement 2*). Importantly, the PICs that display specific Isw2-directed activity are bound by Isw2, while those lacking any detectable nucleosome organization by Isw2 are unbound (*Figure 1A*, middle panel).

It has been shown that Isw2 associates with sequence-specific DNA binding factors, such as the transcriptional repressor Ume6 (*Goldmark et al., 2000*; *Fazzio et al., 2001*). Isw2 activity at Ume6-bound loci has been previously characterized as precise, with Isw2 reproducibly moving nucleosomes until the predicted edge of the nucleosome core particle is 30 base pairs from the center of the Ume6 binding motif (*McKnight et al., 2016*). Because of the connection to Ume6, we examined nucleosome positions in an *isw1/chd1* background in the presence and absence of Isw2 to determine



**Figure 1.** Isw2 is a specialist remodeler that positions single nucleosomes at target sites. (**A**) (Left) Clustered heatmap showing differences in nucleosome dyad signal between *isw2/isw1/chd1* and *ISW2/isw1/chd1* strains at 5942 pre-initiation complex sites (PICs). Black indicates positions where Isw2 preferentially positions nucleosomes compared to the strain lacking Isw2. (Middle) Heatmap of ISW2(K215R) ChIP signal, with rows linked to the PIC data on the left, shows that Isw2-dependent nucleosome changes overlap with regions where Isw2 is present. (Right) Average nucleosome dyad

*Figure 1 continued on next page*

*Figure 1 continued*

signal for wild type (WT) (black), *isw1/chd1* (cyan), and *isw2/isw1/chd1* (red) strains for the 178 PIC sites in cluster 3. Black arrows denote Isw2-driven nucleosome shifts. Green arrows indicate rapid decay of positioning at PIC-distal nucleosomes in the *ISW2/isw1/chd1* mutant. (B) (Left) Genome Browser image showing nucleosome dyad signal at a *u*nscheduled *m*eiotic *g*ene *e*xpression (Ume6) motif (cyan rectangle) for indicated strains. Vertical gray dashed line denotes the motif-proximal WT nucleosome positions while vertical pink dashed line indicates the nucleosome positions in the absence of Ume6 or Isw2. (Center) Clustered heatmap showing the difference in nucleosome dyad signal between *isw2/isw1/chd1* and *ISW2/isw1/chd1* strains at 202 intergenic Ume6 motifs. Black indicates positions where Isw2 preferentially positions nucleosomes compared to strains lacking Isw2. (Right) Average nucleosome dyad signal for indicated strains at Ume6 motifs in cluster 1. Black arrows indicate direction of nucleosome positioning by Isw2. Green arrows signify decreased positioning of motif-distal nucleosomes in the *ISW2/isw1/chd1* strain (cyan) compared to WT (black). (C) (Left) Cartoon depicting the expected activity of Isw2 at barrier elements according to current biochemical data and nucleosome positioning models. Isw2 is thought to move nucleosomes away from bound factors and space nucleosomes with an approximately 200 base pair repeat length. (Right) Cartoon of the observed activity of Isw2 at target sites where only a motif-proximal single nucleosome is precisely positioned but distal nucleosomes are not well-spaced by Isw2.

The online version of this article includes the following figure supplement(s) for figure 1:

**Figure supplement 1.** Isw2 is a precise specialist at target nucleosomes.
**Figure supplement 2.** Isw2 is a precise specialist at target nucleosomes.
**Figure supplement 3.** Isw2 is a precise specialist at target nucleosomes.
**Figure supplement 4.** Isw2 is a precise specialist at target nucleosomes.
**Figure supplement 5.** Isw2 is a precise specialist at target nucleosomes.

whether Isw2 is similarly restricted at known target sites. Again, we determined that Isw2 is efficient at positioning the Ume6-proximal nucleosome but positioning of nucleosomes decays rapidly as the distance from the proximal nucleosome increases, suggesting that Isw2 may only position single nucleosomes at target sites (*Figure 1B*, clusters 1 and 3). Nucleosomes also appear to always be positioned toward Ume6 motifs as nucleosome positions in the absence of Isw2 are always more distal to the Ume6 motif than when Isw2 is present. Finally, these nucleosomes are positioned with the dyad only separated from the Ume6 motif by 100 nucleotides rather than the ~200 nucleotides that would be expected between dyads in a nucleosome array based on Isw2 preferentially leaving 60 base pairs of linker DNA between nucleosomes in vitro (*Kagalwala et al., 2004*; *Tsukiyama et al., 1999*). Of note, a subset of Ume6-bound sites do not display Isw2-dependent nucleosome remodeling (*Figure 1B*, cluster 2). We have observed slightly reduced chromatin immunoprecipitation (ChIP) signal for Ume6 at these sites (*Figure 1—figure supplement 1C*). We speculate that for cluster 2 sites where Ume6 is bound, the Isw2 complex might in fact be recruited but that the nearest nucleosome is too distant from the recruitment site, making it out of reach of the remodeler and thus resulting in no change in nucleosome position at these sites.

The observations that (1) Isw2 is solely required to move single nucleosomes at target sites, (2) Isw2 does not have global nucleosome spacing/organizing activity, and (3) Isw2 moves nucleosomes within 100 nucleotides of bound Ume6 suggest that Isw2 behavior in cells is distinct from our understanding of Isw2 activity from decades of biochemical characterization. Similarly, these specific movements toward Ume6 (a barrier) are inconsistent with previous biophysical studies, where ISWI proteins were shown to move nucleosomes away from inert DNA-bound factors (*Li et al., 2015*). Because of these inconsistencies, we wished to know if Isw2 followed the 'barrier model' for positioning nucleosomes at Ume6-bound targets. To initially test this, we created a variant Ume6 construct where all residues were deleted except for the DNA binding domain. This Ume6(Δ2–763) construct binds to the same targets as full-length Ume6 (*Figure 1—figure supplement 3*). However, Isw2 does not appear to have any activity on global Ume6-proximal nucleosomes in the presence of the Ume6 DNA binding domain alone as nucleosomes in this strain occupy identical positions when Ume6 or Isw2 are completely absent (*Figure 1B*). In the presence of full-length Ume6, the Isw2 complex appears to be necessary and sufficient for moving motif-proximal nucleosomes as nucleosome positions in the *ISW2/isw1/chd1* strain could achieve identical motif-proximal nucleosome positions as the wild-type strain. Additionally, the *CHD1/isw1/isw2* and *ISW1/chd1/isw2* strains were unable to move any Ume6-proximal nucleosomes (*Figure 1—figure supplements 4* and *5*), which strongly argues that Ume6 is not acting as a passive barrier against which nucleosome spacing factors can pack nucleosomes. Instead, these data are more consistent with the recent characterization of Isw2 as a 'puller' (*Kubik et al., 2019*), with Ume6 being a DNA-bound factor that may immobilize Isw2 to

create leverage for 'pulling'. Consistent with this immobilized pulling model and consistent with the directional movement of single nucleosomes toward Ume6-bound sites, artificially tethered chromatin remodeling proteins were previously shown to always move nucleosomes toward target sites (*Donovan et al., 2019*). We suspected that Ume6 and Isw2 likely interact in a specific fashion to faithfully select and precisely move single-target nucleosomes toward a recruitment motif (*Figure 1C*).

## A small helical epitope is necessary and sufficient for Isw2-directed nucleosome positioning at Ume6 targets

To determine which region(s) on Ume6 are required for specific nucleosome positioning by Isw2, we initially created a panel of N-terminal Ume6 truncations to determine when nucleosome positioning by Isw2 is lost (*Figure 2—figure supplement 1*). This initial truncation panel was necessary due to the poor overall conservation of the Ume6 protein even within related yeasts, as well as the disordered structure predicted by Phyre2 (*Kelley et al., 2015*). Our truncation panel indicated that Isw2 activity was retained if the N-terminus was deleted to residue 322 but lost when deleted to residue 508. Closer inspection of the residues between 322 and 508 revealed a conserved region with a proline-rich segment followed by a predicted alpha helix, altogether spanning Ume6 residues 479–508 (*Figure 2A*). Deletion of residues 2–479 preserved Isw2-positioned nucleosomes at Ume6 sites, while an internal deletion of 480–507 in the context of an otherwise full-length Ume6 abrogated nucleosome positioning by Isw2 (*Figure 2A*, *Figure 2—figure supplement 2*). Importantly, Ume6 Δ2–479 and Ume6 Δ2–508 showed identical binding as measured by ChIP (*Figure 2—figure supplement 3*), indicating that the loss of nucleosome positioning is not due to the loss of Ume6 binding.

Since this region is proximal to the characterized Sin3-binding domain in Ume6 (*Washburn and Esposito, 2001*), we wished to validate that the newly determined Isw2-recruitment helix is independent from the Sin3-binding domain. Ume6 recruits both Isw2 and Sin3-Rpd3 for full repression of target genes (*Goldmark et al., 2000*; *Fazzio et al., 2001*). If either Isw2 or Sin3-Rpd3 is present, there is partial repression at Ume6-regulated genes. However, if Sin3-Rpd3 and Isw2 are both lost, Ume6 targets are fully de-repressed. We examined transcriptional output at Ume6 genes in Ume6(Δ2–479) +/− Rpd3 and Ume6(Δ2–508) +/− Rpd3. Transcription was modestly increased at Ume6 targets in Ume6(Δ2–508)/*RPD3+* compared to Ume6(Δ2–479)/*RPD3+* (*Figure 2—figure supplement 4*), which would be expected if only Isw2 is lost when residues 479–508 are deleted. More convincingly, only a modest increase in transcription was seen at Ume6 targets in the Ume6(Δ2–479)/*Δrpd3* strain, suggesting that Isw2 is still present, while the Ume6(Δ2–508)/*Δrpd3* strain displayed extreme induction of Ume6-regulated genes, suggesting that both Isw2 and Rpd3 activity are absent (*Figure 2B*, *Figure 2—figure supplement 5*).

Finally, we wanted to know if the predicted helix consisting of Ume6 residues 479–508 was sufficient to bring Isw2 nucleosome positioning activity to Ume6 target sites. To test this, we employed the SpyCatcher/SpyTag system (*Zakeri et al., 2012*), which creates a spontaneous covalent bond between a short SpyTag peptide and a SpyCatcher domain. We fused the SpyTag peptide to the C-terminus of Ume6(Δ2–596), a construct that is incapable of positioning motif-proximal nucleosomes (*Figure 2—figure supplement 1*). We then appended Ume6 residues 479–508 to the C-terminus of the SpyCatcher domain and introduced this fusion on a yeast expression plasmid driven by the ADH1 promoter. In yeast cells, this would create a fusion protein where the helical element is ectopically displayed on the C-terminus of a DNA binding competent but nucleosome-positioning-deficient construct, connected via a SpyTag-SpyCatcher linker. This fusion protein was capable of fully recapitulating Isw2-positioned nucleosomes at a subset of Ume6 sites (*Figure 2C*, *Figure 2—figure supplements 6* and *7*). Perhaps not surprisingly, considering the non-native positioning of the recruitment helix in this fusion construct, not all Ume6 sites were able to gain proper nucleosome positioning with this chimeric system (*Figure 2—figure supplement 6*). We conclude that the region spanning residues 479–508 in Ume6 is a yeast-conserved Isw2-recruitment domain and is required and sufficient for recruiting Isw2 nucleosome positioning activity to Ume6 targets.



**Figure 2.** A small predicted helix is the Isw2-recruitment epitope in *u*nscheduled *m*eiotic gene *e*xpression (Ume6). (**A**) (Top left) Schematic diagram of Ume6 truncation and deletion constructs used to identify the Isw2-recruitment epitope, with the known Sin3-interacting domain depicted as a green square, the DNA binding domain as a dark blue rectangle, and the putative Isw2-recruitment helix as a light blue rectangle. (Bottom left) Modeled helical peptide (by Phyre2) and sequence conservation of the identified Isw2-recruitment motif in Ume6 constructs from other yeasts. Asterisks denote

*Figure 2 continued on next page*

*Figure 2 continued*

invariant residues. (Right) Nucleosome dyad signal for Ume6 truncation and deletion strains indicates deletion of the region from residues 480 to 507 completely abrogates nucleosome positioning by Isw2 at Ume6 target sites. Vertical dashed gray lines denote wild-type (WT) positions of nucleosomes while vertical dashed pink lines indicate *isw2* or *ume6*-deficient positions of nucleosomes. (B) Genome Browser image showing transcript abundance at three Ume6 target sites for yeast strains lacking Rpd3 with WT Ume6 (gray), Ume6(Δ2–479) (blue), and Ume6(Δ2–508) (orange). Grossly increased transcription is seen when residues 480–507 are deleted, consistent with expected transcriptional increase associated with loss of Isw2 and Rpd3. Upstream repression sequence (URS) sites are indicated as cyan rectangles. No significant increase in transcription is detected when Ume6 residues 2–479 are deleted. Biological replicates are shown to highlight reproducibility. (C) (Top) Cartoon schematic for ectopic display of the Isw2-recruiting helix (residues 480–507) to the C-terminus of a truncated Ume6 construct lacking Isw2-directed nucleosome positioning. A short SpyTag is appended to the C-terminus of the Ume6 construct and residues 480–507 are fused to the SpyCatcher domain and introduced on a yeast expression vector. (Bottom) Nucleosome dyad signal demonstrating recovery of Isw2-directed nucleosome positions at a subset of Ume6 target genes by the ectopically displayed helical element. Vertical dashed gray lines denote WT positions of nucleosomes while vertical dashed pink lines indicate *isw2* or *ume6*-deficient positions of nucleosomes. URS sites are indicated as cyan rectangles. Individual biological replicates for nucleosome positions after ectopic display of the recruitment helix are provided in *Figure 2—figure supplement 4*.

The online version of this article includes the following figure supplement(s) for figure 2:

**Figure supplement 1.** The *u*nscheduled *m*eiotic *g*ene *e*xpression (Ume6) helix between residues 479 and 508 recruits Isw2 to Ume6 targets.

**Figure supplement 2.** The *u*nscheduled *m*eiotic *g*ene *e*xpression (Ume6) helix between residues 479 and 508 recruits Isw2 to Ume6 targets.

**Figure supplement 3.** The *u*nscheduled *m*eiotic *g*ene *e*xpression (Ume6) helix between residues 479 and 508 recruits Isw2 to Ume6 targets.

**Figure supplement 4.** Transcription data support a role of *u*nscheduled *m*eiotic *g*ene *e*xpression (Ume6) residues 479–508 for Isw2 recruitment and not Rpd3 activity.

**Figure supplement 5.** Transcription data support a role of *u*nscheduled *m*eiotic *g*ene *e*xpression (Ume6) residues 479–508 for Isw2 recruitment and not Rpd3 activity.

**Figure supplement 6.** Ectopic display of the *u*nscheduled *m*eiotic *g*ene *e*xpression (Ume6) helical element can rescue Isw2 activity at Ume6 targets.

**Figure supplement 7.** Ectopic display of the *u*nscheduled *m*eiotic *g*ene *e*xpression (Ume6) helical element can rescue Isw2 activity at Ume6 targets.

## A similar helical element exists in Swi6, a newly identified Isw2-recruitment adapter protein

While dissecting the Isw2-recruitment domain in Ume6, we discovered that deleting the *MBP1* gene resulted in ectopic nucleosome positioning at a subset of Mbp1 target loci, which was identical to mispositioned nucleosomes in a Δ*isw2* strain. Mbp1 is a conserved cell cycle regulator that complexes with Swi6 to form the MBF complex (*Koch et al., 1993*). This complex activates the transition from G1 to S and includes the conserved function of regulating Start-specific transcription (*Koch et al., 1993*; *Breeden, 1996*). To determine how Mbp1 recruits Isw2, we similarly made truncations of Mbp1 to determine at which point nucleosome positioning no longer resembles wild-type positioning and reflects Δ*isw2* positioning instead. The DNA binding element in Mbp1 resides in the extreme N-terminus (*Figure 3A*) spanning residues 2–124 (*Nair et al., 2003*), so a panel of C-terminal truncations was created. However, before examining the full panel of truncations, we observed that nucleosome positioning was already identical to Δ*isw2* positioning in Mbp1 Δ562–833, the first C-terminal truncation examined (*Figure 3—figure supplement 1*). This extreme C-terminal region interacts with Swi6 (*Figure 3A*), so we speculated that Swi6 may be responsible for recruiting Isw2. As predicted, deletion of the *SWI6* gene led to ectopic nucleosome positions identical to Δ*mbp1* and Δ*isw2* strains at the small subset of Mbp1 targets.

We conducted sequence alignment and conservation analyses between the helical element in Ume6 and full-length Swi6 from multiple yeast species (*Figure 3A*). We noticed a similarly conserved surface-exposed helix (*Foord et al., 1999*) in the cell cycle regulating protein Swi6 (*Figure 3A*). Intriguingly, the function of this helical element has not been determined despite its sequence conservation. Because Swi6 also interacts with Swi4 to form the highly conserved SBF complex (*Koch et al., 1993*), we speculated that deletion of either Swi6, Swi4, or Isw2 could potentially lead to ectopic nucleosome positions at a subset of SBF targets. Indeed, we observed ectopic nucleosome positioning at the HSP12 locus (an SBF target) when either Isw2, Swi6, or Swi4 was absent (*Figure 3A*). Wild-type nucleosome positions were observed in the absence of Mbp1, indicating that this is specific to SBF. Similarly, wild-type nucleosome positions were observed at Mbp1 targets when Swi4 was missing (*Figure 3A*), again suggesting that MBF and SBF have individual Isw2-targeting capacity at their respective binding sites. Swi6 appears to be an adapter protein responsible for recruiting Isw2 to Mbp1 and Swi4 sites since Swi6 has no intrinsic DNA binding domain.

**Figure 3.** The cell cycle regulator Swi6 contains a similar helical element and recruits Isw2 to MBF and SBF target genes. (**A**) (Top left) Schematic representation of the Swi6-Mbp1 MBF complex. Swi6 interacts with Mbp1 through the C-terminal domain (black rectangle). Mbp1 has an N-terminal DNA binding domain (dark blue rectangle). The putative Isw2-recruitment helix is in the Swi6 N-terminus (light blue rectangle). (Center left) Conserved residues in the putative Isw2-recruitment helix in Swi6 for three yeast species compared to the Isw2-recruitment helix in Ume6 for *S. cerevisiae*. (Bottom

*Figure 3 continued on next page*

*Figure 3 continued*

left) Crystal structure (Protein Data Bank [PDB] ID 1sw6) showing the location of the surface-exposed, conserved helical element from Swi6 in red. (Top right) Schematic representation of the Swi6-Swi4 SBF complex. Swi6 interacts with Swi4 through the C-terminal domain (black rectangle). Swi4 has an N-terminal DNA binding domain (green rectangle). Putative Isw2-recruitment helix is shown (small blue rectangle). (Bottom center) Genome Browser image showing nucleosome dyad signal for indicated strains at the *RAD51* locus, an MBF target gene with an indicated Mbp1 binding motif (blue rectangle). Wild-type (WT) nucleosome positions are indicated by vertical dashed gray lines while ectopic positions associated with *isw2*, *mbp1*, and *swi6* deletion strains are indicated by vertical dashed pink lines. (Bottom right) Genome Browser image showing nucleosome dyad signal for indicated strains at the *HSP12* locus, an SBF target gene with an indicated Swi4 binding motif (green rectangle). WT positions are denoted by vertical gray dashed lines while ectopic nucleosome positions associated with *isw2*, *swi6*, and *swi4* deletion strains are indicated with vertical pink dashed lines. (B) (Top) Schematic representation of constructs used to determine if ectopic display of an Isw2-recruitment helix on the Mbp1 N-terminus could recover Isw2-positioned nucleosomes at Mbp1 target genes. Either WT Mbp1, a C-terminal deletion of Mbp1 leaving only the DNA binding domain and an appended SpyTag, or a C-terminal deletion of Mbp1 leaving the DNA binding domain and SpyTag with constitutively expressed SpyCatcher fused to the Isw2-recruitment helix from Ume6 was examined. (Bottom) Genome Browser image showing nucleosome dyad signal for indicated strains at the *ESC8* (left) or *RAD51* (right) loci. Gray vertical dashed lines indicate WT nucleosome positions while vertical dashed pink lines indicate ectopic nucleosome positions associated with inactive Isw2 or Mbp1/Swi6. Biological replicates for ectopic display of the recruitment helix are provided as two separate tracks (gold) to emphasize reproducibility.

The online version of this article includes the following figure supplement(s) for figure 3:

**Figure supplement 1.** Truncation of the Mbp1 C-terminus eliminates Isw2-directed nucleosome positioning at Mbp1 targets.

To determine if Isw2 recruitment to Mbp1 sites was sufficient to recapitulate proper nucleosome positioning, we again used a SpyTag-SpyCatcher approach (*Figure 3B*). Mbp1 was truncated to the DNA binding domain alone (Mbp1 1–136), which abolishes its interaction with Swi6 but still allows for proper genomic localization. This truncation construct was appended with SpyTag, and nucleosome positions were examined in the absence of any SpyCatcher partner present. As expected, we observed aberrant chromatin structure identical to the Δ*isw2* strain near the Isw2-dependent Mbp1 targets, adjacent to Mbp1 consensus motifs (*Figure 3B*). We then introduced SpyCatcher fused to the helical element from Ume6, which was characterized above for bringing Isw2 to Ume6-bound loci. Introduction of the SpyCatcher-Ume6 fusion to the Mbp1(1–136)-SpyTag background resulted in the rescue of proper Isw2-directed nucleosome positioning at Mbp1 sites (*Figure 3B*). Altogether, these data strongly support our model that these conserved, putatively helical sequences are important for recruiting Isw2 to establish proper chromatin structure at multiple sequence-specific motifs throughout the genome. We also implicate Swi6 as an adapter protein for bringing Isw2 to a small subset of both Swi4 and Mbp1 targets to create Isw2-specific nucleosome positioning at these genes. Finally, the ectopic display of an Isw2-recruitment helix can recapitulate proper Isw2-directed nucleosome positioning, further supporting the notion that a small epitope is necessary and sufficient for communicating specific nucleosome positioning outputs to the Isw2 chromatin remodeling protein.

## The conserved WAC domain in Itc1 is the targeting domain of the Isw2 complex

The Isw2 complex contains two major subunits (*Figure 4A*). The catalytic subunit Isw2 harbors the energy-producing ATPase domain flanked by biochemically well-defined autoregulatory domains (*Clapier and Cairns, 2012*; *Yan et al., 2016*; *Ludwigsen et al., 2017*) with a C-terminal HAND-SANT-SLIDE domain, thought to bind linker DNA (*Zofall et al., 2004*) and interact with the accessory subunit Itc1. Itc1 contains an N-terminal WAC domain, thought to bind to and sense extranucleosomal DNA and help with nucleosome assembly in the *Drosophila* ortholog ACF1 (*Fyodorov and Kadonaga, 2002*). Itc1 links to Isw2 through a DDT domain (DDT is named for "DNA-binding homeobox-containing proteins and different transcription factors") (*Fyodorov and Kadonaga, 2002*). The ~350 amino acid N-terminal region of human Acf1 was shown to bind both extranucleosomal linker DNA and the histone H4 tail, suggesting an allosteric mechanism through which ISWI complexes can set proper spacing between nucleosomes (*Hwang et al., 2014*). Though this work was performed with human ACF complex, Hwang et al. demonstrated that removal of residues 2–374 in *S. cerevisiae* was lethal, suggesting a critical and conserved role of these residues in establishing proper chromatin structure in vivo (*Hwang et al., 2014*).



**Figure 4.** The N-terminal WAC domain in Itc1 couples Isw2 biochemical activity to all Isw2 genomic targets. (**A**) Cartoon representation the Isw2 and Itc1 subunits of the yeast ISW2 complex. Isw2 possesses autoregulatory domains on either side of the catalytic ATPase domain (AutoN and NegC). The HAND-SANT-SLIDE (HSS) domain of Isw2 interacts with the DDT domain of Itc1 for complex formation. Itc1 has an N-terminal region thought to act

*Figure 4 continued on next page*

*Figure 4 continued*

as a length-sensing domain (gray rectangle) and an N-terminal WAC domain with putative nonspecific linker DNA binding ability. (**B**) Nucleosome sliding assay demonstrating that deletion of the N-terminal domain (Δ9–374) from Itc1 does not impair nucleosome sliding in vitro by the Isw2 complex. Higher electrophoretic mobility indicates end-positioned (unslid) nucleosomes while lower electrophoretic mobility indicates centrally positioned (slid) nucleosomes. Isw2-FLAG complexes were purified from exponentially growing yeast cells as described in 'Materials and methods'. Amount of Isw2 added was 1 μl (+) or 1.5 μl (++). Sliding assays were performed three independent times with similar results. (**C**) Genome Browser images showing nucleosome dyad positions for indicated strains at *RAD51* and *ALP1*, two representative Isw2 targets. Only wild-type (WT) cells display the proper nucleosome positions (vertical gray dashed lines) while all Itc1 truncations and *isw2* deletion display similar ectopic nucleosome positions (vertical pink dashed lines). (**D**) Heatmap comparing difference in nucleosome positions at 5942 PIC locations for *isw2* deletion versus WT strains (left) and Itc1(Δ24–130) versus WT strains (right). Black indicates where nucleosomes are shifted by functional Isw2 while red indicates where nucleosomes shift when Isw2 complex is perturbed. All rows are linked and ordered identically to *Figure 1A*.

Because of the geometry of the Isw2 complex, with the N-terminus of Itc1 sensing DNA information distal to the nucleosome onto which the catalytic subunit is engaged, we speculated that the N-terminus of Itc1 would be the most likely component of the Isw2 complex for interacting with epitopes in DNA-bound recruitment factors. We first attempted to recapitulate the result from Hwang et al. and made the identical Itc1(Δ2–374) deletion. Isw2 containing Itc1(Δ2–374) did not display any defects in nucleosome sliding using a gel mobility shift assay that detects nucleosome centering by Isw2 (*Figure 4B*). Surprisingly, this construct was not lethal in our W303 background, but phenocopied a Δ*isw2* strain by displaying identical ectopic nucleosome positioning at all Isw2 target sites throughout the genome (*Figure 4C*). Since proper targeted nucleosome positioning was lost when this large N-terminal region was removed, but complex formation and catalytic activity were maintained, we strongly suspected that the Isw2 targeting domain resided in the Itc1 N-terminus. We created a panel of truncations in this region, guided by sequence conservation through humans, and determined whether wild-type or Δ*isw2* positions were observed throughout the genome. All truncations tested resulted in loss of positioning at Isw2 targets, and we were able to narrow the targeting region entirely to the highly conserved WAC domain. Deletion of the WAC domain (Itc1 residues 24–130) produced identically ectopic nucleosome positions compared to Δ*isw2* at target loci (*Figure 4C*) and genome-wide (*Figure 4D*). We conclude that the WAC domain of Itc1 is the component of the Isw2 complex responsible for coupling with epitopes on DNA-bound factors such as Ume6, Swi6, and all other Isw2 targeting proteins with yet-to-be-defined recruitment epitopes.

## The WAC domain binds Isw2 targets and orients the catalytic subunit on target-proximal nucleosomes

To confirm that the WAC domain can interact with Isw2 targets throughout the genome, we created Itc1(1-73)-FLAG and Itc1(1-132)-FLAG constructs based on two differentially conserved regions within the full WAC domain (*Figure 5A*). Neither of these constructs contains the DDT domain, so they are incapable of forming a complex with endogenous Isw2. We performed ChIP-Seq to determine if these WAC domain constructs could associate with Isw2 targets without complexing with the Isw2 catalytic domain (*Figure 5B*, *Figure 5—figure supplement 1*). Genome-wide binding demonstrates large, but not complete overlap of Isw2(K215R)-FLAG ChIP peaks with both Itc1(1–73)-FLAG and Itc1(1–132)-FLAG, strongly suggesting that the Itc1 region from 1 to 73 alone can interact with Isw2 targets.

We noticed that the Itc1 signal and Isw2 signal were offset at target genes such that Itc1(1–73) or Itc1(1–132) was upstream and Isw2 was closer to the nucleosome that was selected for repositioning (*Figure 5B*, *Figure 5—figure supplement 1*). Genome-wide analysis showed that Itc1(1–73) was associated with approximately half of Isw2-bound loci and was offset from the catalytic subunit at all co-bound sites (*Figure 5C*). In all cases, Itc1(1–73) was found upstream of the nucleosome that was repositioned, and Isw2 was located on top of the selected nucleosome. Nucleosomes were always shifted toward the Itc1 subunit (*Figure 5C*). This geometry matches what was seen by ChIP-Exo mapping with Isw2 subunits at Reb1 target sites (*Yen et al., 2012*). We propose a mechanism where the Itc1 WAC domain interacts with a DNA-bound factor, which constrains the Isw2 catalytic subunit



**Figure 5.** The Itc1 WAC domain associates with genomic Isw2 targets and orients Isw2 on the proper nucleosomes. (**A**) Sequence conservation for regions of Itc1 examined by ChIP. Itc1(1–73)-FLAG incorporates the pink highlighted region while Itc1(1–132)-FLAG incorporates the pink and green highlighted regions. Sequence conservation is shown relative to human BAZ1A and *Drosophila melanogaster* Acf1, two widely studied Itc1 orthologs. (**B**) (Left) Full view of yeast chromosome IX showing Isw2(K215R)-FLAG ChIP (red), Itc1(1–73)-FLAG ChIP (pink), Itc1(1–132)-FLAG ChIP (green),

*Figure 5 continued on next page*

*Figure 5 continued*

nucleosome dyad signal from wild-type (WT) yeast (black), and nucleosome dyad signal from Itc1(Δ24–130) yeast (blue). Regions indicated by black rectangles are shown with higher resolution on the right. (Right) Zoomed-in view of a locus where Isw2-ChIP and Itc1 truncation ChIP overlap (*RGI2*) or where only Isw2 binding is detected (*PDR11*). Black circles indicate center of ChIP peaks and are connected by a dashed black line to highlight offset of indicated peaks. (C) (Left) Heatmap showing 273 detected Isw2 ChIP peaks (red) clustered by associated Itc1(1–73)-FLAG ChIP (pink). The two clusters (right-side Itc1 and left-side Itc1) are shown on the right. (Right) Meta-analysis of Isw2(K215R)-FLAG ChIP signal at 62 cluster 1 peaks or 57 cluster 2 peaks (from left) with associated Itc1(1–73)-FLAG signal. The offset between Isw2 and Itc1 is indicated by two circles connected by a dashed line. Associated nucleosome positions for WT and *isw2* deletion strains for each cluster are shown below in black and blue, respectively. All data are centered at called Isw2 peaks. (D) Cartoon representation for how the N-terminal WAC domain of Itc1 interacts with a helical element in a sequence-specific DNA-associated transcription factor to orient Isw2 on the proper motif-proximal nucleosome for directional movement toward the recruitment site.

The online version of this article includes the following figure supplement(s) for figure 5:

**Figure supplement 1.** The WAC domain orients the Isw2 catalytic domain at nearly half of detected Isw2 targets in yeast.

to select the proper proximal nucleosome and reposition it toward the immobilized Itc1 (*Figure 5D*). This is again consistent with the recently proposed 'pulling' model (*Kubik et al., 2019*), but we postulate that Itc1 is anchored to a DNA-bound factor such as Ume6 to allow Isw2 to pull nucleosomes toward the proper location.

## Essential acidic residues required for targeting are lost in the *Drosophila* genus, explaining biochemical and genetic inconsistencies

There is an abundance of literature suggesting that *Drosophila* ACF complex, the Isw2 ortholog, is a nonspecific nucleosome spacing and assembly factor that evenly spaces phased nucleosome arrays against defined genomic barriers (*Lusser et al., 2005*; *Baldi et al., 2018*; *Fyodorov and Kadonaga, 2002*). We wondered if the WAC domain of *Drosophila* Acf1 was different from that of Itc1, so we performed sequence alignment of WAC domains and compared to Acf1 from the *Drosophila* genus. While sequence alignment demonstrated widespread conservation of the WAC domain, one striking feature was exposed: the *Drosophila* genus underwent reversal or loss of negative charge at multiple residues that are strictly or mostly acidic in other representative organisms (*Figure 6A*).

Two of these residues are strictly acidic in all organisms except members of the *Drosophila* genus (E33 and E40 in Itc1). The other two (E43 and D70 in Itc1) are more loosely conserved, though they are strictly positive charge in *Drosophila*. We made charge-reversal mutations in *S. cerevisiae* Itc1 to recapitulate the *D. melanogaster* residues at each of these positions either pairwise (a, b and c, d to separate the strictly conserved acidic versus loosely conserved acidic nature) or simultaneously (a, b, c, d) to reverse all charges to the *D. melanogaster* sequence. We assessed whether charge reversal was sufficient to abrogate targeted nucleosome positioning at Isw2 targets across the yeast genome (*Figure 6B*). Strikingly, the E33R/E40H double mutation (a, b) was enough to completely abolish Isw2 activity at specific and known Isw2 targets (*Figure 6B*) and at all genomic loci where Isw2 activity is observed (*Figure 6C*). Mutation of the less-conserved acidic residues E43R/D70K (c, d) retained Isw2-directed nucleosome positioning. As expected, mutation of all four acidic residues (a, b, c, d) E33R/E40H/E43R/D70K resulted in complete loss of Isw2-targeted activity across the genome (*Figure 6B, C*). We conclude that the *Drosophila* genus lost critical acidic residues that are essential for targeted nucleosome positioning by *S. cerevisiae* Isw2, potentially explaining the disconnect between the *Drosophila* ACF literature and what we have characterized herein. It is possible that the increase in positive charge simultaneously increases nonspecific binding of *Drosophila* Acf1 to extra-nucleosomal DNA, and these charge reversals may help explain the nonspecific spacing behavior of Acf1 observed in *Drosophila*. We also believe that there is strong potential that humans and most other organisms have retained targeting potential as they retain mechanistically important acidic residues present in yeast Itc1. In support of conservation, targeted nucleosome array formation has previously been observed in humans at specific transcription factor sites including CTCF, JUN, and RFX5 (*Wiechens et al., 2016*).



**Figure 6.** Essential targeting-specific charged residues in the conserved WAC domain is lost in *Drosophila*. (A) Sequence conservation of the N-terminal region (22–73) of Itc1 across various organisms with key charged residues highlighted in blue or red for positive and negative charge, respectively. Horizontal dashed line indicates separation of all other species from members of the *Drosophila* genus. (B) Genome Browser image showing nucleosome dyad signal at two representative Isw2 target loci. Wild-type (WT) nucleosome positions are indicated by gray vertical dashed lines while ectopic nucleosome positions associated with *isw2* deletion or indicated charge reversal mutations are denoted by vertical pink dashed lines. (C) Meta-analysis of nucleosome dyad signal at 178 PIC sites associated with cluster 3 (from *Figure 1A*). Only WT and charge reversal c, d display proper nucleosome positions while charge reversal a, b or a, b, c, and d display ectopic positions identical to deletion of *ISW2* completely.

## Discussion

### An interacting barrier model for nucleosome array establishment

Collectively, our results give rise to an 'interacting barrier model' as an alternative means of genomic nucleosome positioning by introducing a targeted interaction between an epitope contained within condition-specific transcription factors and ISWI-type ChRPs (*Figure 7*). We show that a recruitment

## Obligate Targeted Activity in Cells

**A**

Active Isw2 Complex

WAC Domain

Isw2

Itc1

Recruitment Factor

Requires Acidic Residues

**B**

Strained Isw2 Complex

Shifted Nucleosome

**C**

Precise Distance

Released Isw2 Complex

**Figure 7.** The interacting barrier model for specific nucleosome placement by Isw2. (**A**) A DNA binding factor with an Isw2-recruitment helix (or other epitope) associates with DNA. The WAC domain of Itc1 engages with the recruitment epitope to proximally align the catalytic subunit of Isw2 with the proper nucleosome. (**B**) Nucleosome sliding by Isw2 creates a nucleosome position that is too close to the recruitment epitope for proper alignment of the recruitment epitope – Itc1 WAC – Isw2 catalytic subunit axis, leading to a 'strained complex'. (**C**) The precise distance between the DNA-bound Isw2-recruitment factor and proximal nucleosome after nucleosome positioning by Isw2 is no longer a good substrate for Itc1 WAC interaction and further remodeling, so the Isw2 complex diffuses to new target loci.

factor, the sequence-specific repressor Ume6, harbors a helical domain that interacts with the N-terminus of the Isw2 accessory protein Itc1. Further, we reveal this geometrically restricts the binding of the Isw2 catalytic subunit to a motif-proximal nucleosome. The complex then remodels the nucleosome, repositioning it to a specific distance from the Ume6 recognition motif. At this point and for reasons to be elucidated, this complex is strained or inactivated, and it fails to remodel any further, leaving the nucleosome in a precise location with respect to the bound recruitment factor. The activity of Isw2 and the interacting barrier sets the absolute phase of a nucleosome array that is propagated by true nonspecific spacing activities of Chd1 and Isw1 in yeast, as previously described (*Gkikopoulos et al., 2011*; *Zhang et al., 2011*; *Ocampo et al., 2016*). This 'interacting barrier model' of chromatin organization is more comparable to the factor-targeted activities of SWI/SNF than the nonspecific array spacing of CHD family remodelers and is potentially conserved through humans based on conservation of key interacting residues in Itc1 (*Figure 6A*) and the observation that Isw2 orthologs can precisely position nucleosomes adjacent to specific factors in the human genome (*Wiechens et al., 2016*). Together, we show that coupling between an epitope on an interacting barrier and a conserved chromatin remodeling protein leads to robust, directional, and specific nucleosome organization at genomic regulatory elements.

## Small epitopes in transcription factors organize large chromatin domains

Our data suggest that some small peptide domains embedded within transcription factors can nucleate nucleosome arrays of over 1 kb in length in vivo through an interaction with evolutionarily conserved ChRPs. Unlike the arrays established by nonspecific ChRPs, these nucleosome arrays are organized in a sequence-specific and directional manner. Establishing large swaths of chromatin structure by appending a small epitope on a genome-associated protein creates opportunity for diversity with few evolutionary constraints. Only changes in relatively small DNA binding motifs and the small peptide sequences with which they interact can have a large impact on chromatin structure. Supporting this notion, we were able to identify a strikingly similar motif to that found in Ume6 in the unrelated cell cycle regulator Swi6, which we identified as a new Isw2-recruitment adapter for Swi4 and Mbp1. For these reasons, we find it likely that more ChRP-interacting motifs will be discovered in multiple transcription factors from a variety of organisms, and these motifs may play a significant role in sequence-specific nucleosome positioning for precisely phased and tunable nucleosome arrays in eukaryotic genomes. Importantly, the identification of such epitopes in human cells could lead to the development of targeted drugs to specifically disrupt defined remodeler–transcription regulator interactions.

## Isw2 is obligately targeted to specific nucleosomes without global spacing activity

We found that Isw2 acts on specific targets through these specific transcription factor interactions rather than acting on all nucleosomes genome-wide. We therefore speculate that Isw2 is in a globally repressed state in cells and activated solely on target nucleosomes. This inactivity is not consistent with work in vitro and may be caused by a regulatory interaction that has not been previously observed in biochemical systems. For example, an unknown inhibitory factor that interacts with Isw2 or the nucleosome in cells may be lost during protein purification, allowing for the ubiquitous Isw2 chromatin remodeling activity observed in vitro. Additionally, it is conceivable that Isw2 is unable to bind to linker DNA the same way in a genomic context as it can bind in vitro, potentially due to the presence of unknown chromatin interacting components, molecular crowding, chromatin folding, or other physiological differences not recapitulated in vitro. Maintaining Isw2 in an inactive state may allow organisms to conserve energy by controlling errant ATP hydrolysis while simultaneously enabling for rapid changes in chromatin structure and cellular output in differing contexts. It will be of great interest to determine how interactions with recruitment factor epitopes may alter the activity of Isw2 to elicit such precise nucleosome positioning outcomes in a cellular context.

## The conserved WAC domain tethers Isw2 to transcription factor epitopes in *S. cerevisiae*

The WAC domain, a broad N-terminal region of the Itc1 accessory protein, has been previously characterized as a DNA binding element that shares sequence conservation from flies to humans (*Fyodorov and Kadonaga, 2002*; *Ito et al., 1999*). In this work, we have identified a previously undefined function of the WAC domain in mediating protein–protein interactions between ChRPs and transcription factors in vivo. We have further demonstrated that this mediation requires two conserved acidic residues within the WAC domain, which may allow future work to distinguish the DNA and protein binding capacity of the broader WAC domain region. Intriguingly, these critical acidic residues that are conserved between yeast, humans, mice, and fish have undergone an evolutionary charge reversal in *Drosophila*. It is conceivable that this charge reversal establishes a more general role in generating chromatin structure for the single ISWI-type protein found in flies, as opposed to the more specialized and context-dependent roles of the many ISWI-type ChRPs found in other organisms.

A recent model suggested strong interplay between the human Acf1 (yeast Itc1) N-terminus, extranucleosomal DNA, and the histone H4 tail (*Hwang et al., 2014*). In this model, the human Acf1 N-terminus binds to extranucleosomal DNA in nucleosomes with long linker length, allowing the Snf2h (Isw2) catalytic subunit to engage the H4 tail. Snf2h engagement of the H4 tail relieves known autoinhibitory interactions (*Clapier and Cairns, 2012*; *Ludwigsen et al., 2017*), thereby activating the remodeling complex. When linker DNA length shortens, the N-terminus of Acf1 switches to binding the H4 tail, thus displacing the Snf2h catalytic subunit and inactivating the complex through autoinhibition. This model was used to describe how ISWI complexes can be allosterically inactivated when linker DNA length shortens on nucleosomes and is a mechanistic model for how nucleosome length sensing can be achieved.

Our results indicate that the N-terminus of Itc1 does not have a primary cellular function of length sensing and nucleosome spacing. If the Itc1 N-terminus can bind H4 tail and transcription factor epitopes similarly to extranucleosomal DNA and H4 tail, the Hwang et al. model (*Hwang et al., 2014*) can mechanistically explain the precise distance measurements made at targeted sites in cells. In this speculative model, Itc1 binds a targeting epitope at a genomic locus when the upstream nucleosome is far away. This orients the catalytic subunit on the appropriate nucleosome, which is remodeled toward the recruitment site. When the length between the nucleosome and the recruiting epitope is short enough, the Itc1 N-terminus may bind the H4 tail to inactivate Isw2 through autoinhibition. This function of binding the positively charged H4 tail may be facilitated by the clustered acidic residues in the WAC domain or may be mediated by another domain within the broadly defined 374 base pair Itc1 N- terminus implicated in putative H4 tail binding. Determining whether this interplay between a transcription factor epitope and the H4 tail can tune distance measurements in cells will be important in future biophysical characterizations.

## The benefits of epitope-mediated chromatin remodeling and an interacting barrier

What advantage may an interacting barrier provide that a general barrier cannot, particularly since a noninteracting barrier can still phase nucleosome arrays? We envision at least two major advantages of the interacting barrier model. First, an interacting barrier can behave directionally while a noninteracting barrier cannot. Indeed, Isw2 seems to be positioned on a specific barrier-proximal nucleosome through interactions between the WAC domain of Itc1 and the epitope on Ume6 or Swi6. Directionality allows for more refined establishment of transcriptionally relevant chromatin arrays. Second, an interacting barrier can be modulated in a condition-specific manner through post-translational modification of the small epitope on the Isw2-recruitment factor. For example, one of the proteins that we identified as containing an Isw2- recruitment helix in yeast is Swi6, a critical regulator of the cell cycle in the G1/S transition. Interestingly, only three Swi6-regulated genes were identified as Swi6-mediated Isw2-recruitment sites. It is thus likely that the Swi6–Isw2 interaction can be tuned by cellular context, which is not possible for noninteracting barriers. Importantly, a tunable interacting barrier allows for continuous expression of the barrier and the ability to alter its barrier activity. This is a versatile mechanism through which chromatin structure may be spatiotemporally regulated in a dynamic fashion through these ChRP-recruitment factor interactions. We predict that other targeting

factors likely exist, which might recruit the Isw2 complex to specific sites in the genome through a variety of different domains, and that this creates a mechanism by which remodeling occurs at specific subsets of genomic loci in response to the presence of these recruitment factors. The expression level of transcription factors is controlled by cellular context, such as cell cycle progression or stress response, and in this way the recruitment of Isw2 to specific sites might also be linked to changes in cell conditions.

# Materials and methods

## Key resources table

| Reagent type (species) or resource | Designation | Source or reference | Identifiers | Additional information |
|---|---|---|---|---|
| Strain, strain background (*Saccharomyces cerevisiae*) | W303-1A | Laboratory of Rodney Rothstein | American Type Culture Collection (ATCC): 208352 | |
| Strain, strain background (*S. cerevisiae*) | | | | For additional strains, see *Supplementary file 1* |
| Antibody | Anti-FLAG M2 (mouse monoclonal) | Sigma | Cat# F1084; RRID:AB_262044 | 5 µl antibody per 30 µl beads |
| Antibody | Anti-FLAG M2 magnetic beads (mouse monoclonal) | Sigma | Cat# M8823; RRID:AB_2637089 | |
| Recombinant DNA reagent | Plasmid | | | For plasmid informatio,n see *Supplementary file 2* |
| Peptide, recombinant protein | Dynabeads Protein G | Thermo Fisher Scientific | Cat# 10004D | 200 µl |
| Peptide, recombinant protein | 3x FLAG | Sigma | Cat# F4799-4MG | MDYKDHDGDY KDHDIDYKDDDDK |
| Commercial assay or kit | MinElute PCR purification kit | Qiagen | Cat# 28006 | |
| Commercial assay or kit | Ovation Ultralow System V2 | NuGEN | Cat# 0344NB-32 | |
| Commercial assay or kit | Universal Plus mRNA-Seq | NuGEN | Cat# 0508-32 | |
| Commercial assay or kit | Quant-IT PicoGreen dsDNA assay kit | Invitrogen | Cat# P7589 | |
| Chemical compound, drug | Proteoloc protease inhibitor cocktail (EDTA-free) | Expedeon | Cat# 44214 | |
| Chemical compound, drug | Nuclease, micrococcal | Worthington Biochemical | Cat# LS004798 | |
| Chemical compound, drug | Zymolyase (R) 100T | AMS Bio | Cat# 120493-1 | |

*Continued on next page*

*Continued*

| Reagent type (species) or resource | Designation | Source or reference | Identifiers | Additional information |
|---|---|---|---|---|
| Chemical compound, drug | Agencourt AMPure XP beads | Beckman Coulter | Cat# A63881 | |
| Chemical compound, drug | Oligo(dT)$_{25}$ beads | NEB | Cat# S1408S | |
| Software, algorithm | Trimmomatic | PubMed ID (PMID):24695404 | RRID:SCR_011848 | |
| Software, algorithm | STAR (V.2.5.3) | PMID:23104886 | RRID:SCR_015899 | |
| Software, algorithm | HTSeq (V.0.9.1) | PMID:25260700 | RRID:SCR_005514 | |
| Software, algorithm | DESeq2 (V.1.22.2) | PMID:25516281 | RRID:SCR_015687 | |
| Software, algorithm | ggplot2 | ISBN 978-0-387-98140-6 | RRID:SCR_014601 | |
| Software, algorithm | Bowtie 2 | PMID:22388286 | RRID:SCR_005476 | |
| Software, algorithm | Integrated Genome Browser | PMID:27153568 | RRID:SCR_011792 | |

## Yeast strains and plasmids

All yeast strains were derived from the parent strain *S. cerevisiae* W303 RAD5+. Gene deletions were made by replacing the gene of interest with antibiotic resistance markers amplified from pAG vectors. C-terminal deletions of genes were also made by replacing the region to be deleted with antibiotic resistance markers. N-terminal gene deletions were made by first replacing the region to be deleted with a URA3 marker, and then counterselecting with 5-fluoroorotic acid (FOA) to delete the URA3. Ume6-helix was introduced to yeast through plasmid transformation of a p416 vector containing the Ume6 helix fused to the SpyCatcher protein (*Zakeri et al., 2012*). To make SpyTagged yeast strains, a C-terminal 3x FLAG tag followed by the SpyTag sequence (AHIVMVDAYKPTK) (*Zakeri et al., 2012*) was cloned into a pFA6a vector. Tags were then inserted at the endogenous locus of interest by homologous recombination of PCR products from the respective tagging vectors using selectable drug markers.

## Growth conditions

Cells were grown at 30°C and 160 rpm in yeast extract–peptone–2% glucose (YPD) medium unless otherwise indicated. Strains were streaked from glycerol stocks onto 2% agar YPD plates and grown at 30°C for 2–3 days. An isolated colony was then grown overnight in 25 ml of YPD. This pre-culture was used to inoculate 25 ml of YPD at an $OD_{600}$ of 0.2, which was grown to an $OD_{600}$ of 0.6–0.8 for chromatin analysis. Yeast containing nonintegrating plasmids (p416) were grown in SD (-)Ura overnight, diluted to $OD_{600}$ = 0.2 in YPD and grown to $OD_{600}$ = 0.6–0.8 for chromatin analysis. Cells were then fixed with 1% formaldehyde and harvested for chromatin analysis.

## Protein purification

Yeast strains containing the Isw2 variants of interest appended with a FLAG tag were grown at 30°C to an $OD_{600}$ of ~1. Yeast were pelleted, washed with binding buffer (25 mM HEPES, pH 7.8, 300 mM NaCl, 0.5 mM EGTA [ethylene glycol-bis(β-aminoethyl ether)-N,N,N',N'-tetraacetic acid], 0.1 mM EDTA [ethylenediamine tetraacetic acid], 2 mM MgCl$_2$, 20% glycerol, 0.02% NP-40, 2 mM beta-mercaptoethanol, 1 mM PMSF [phenylmethylsulfonyl fluoride], 1× protease inhibitor cocktail [Expedeon, Cambridge UK]), and then lysed via cryogrinding. Yeast powder was incubated with binding buffer for 90 min before the addition of 200 µl bed volume anti-FLAG magnetic beads (Sigma M2; Sigma, St. Louis MO USA). After 3 hr incubation at 4°C, beads were collected with

magnets and washed three times with binding buffer and three times with elution buffer (25 mM HEPES pH 7.8, 500 mM NaCl, 0.5 mM EGTA, 0.1 mM EDTA, 2 mM MgCl₂, 20% glycerol, 0.02% NP-40, 2 mM beta-mercaptoethanol, 1 mM PMSF, 1× protease inhibitor cocktail [Expedeon, Cambridge UK]). A 0.5 mg/ml solution of FLAG peptide in 100 ml elution buffer was then added to the beads and allowed to incubate for 30 min. This process was repeated three more times for a total of four elutions. Elutions were analyzed by silver staining and combined by estimated purity for aliquoting and storage at −80°.

## Nucleosome sliding assay

Sliding assays were performed at least three independent times with reproducible results.

Recombinant yeast histones were purified as previously described (*Luger et al., 1999*) and dialyzed by gradient salt dialysis onto the Widom 601 positioning sequence to create end-positioned nucleosomes with 60 base pairs of linker DNA (*Lowary and Widom, 1998*). Nucleosome sliding was performed at 25°C in sliding buffer (50 mM KCl, 15 mM HEPES, pH 7.8, 10 mM MgCl₂, 0.1 mM EDTA, 5% sucrose, 0.2 mg/ml bovine serum albumin, with or without 5 mM ATP) by incubating 1 ml or 1.5 ml of purified Isw2 with 12.5 nM reconstituted mononucleosomes for 40 min in 6 ml reaction volume. Reactions were quenched by diluting 1:2 with solution containing 3 mM competitor DNA and 5% sucrose. Native PAGE (6%) was used to separate the positioning of the mononucleosomes, with Cy5.5-labeled nucleosomal DNA detected by a LiCor Odyssey FC imager (LI-COR Biosciences, Lincoln NE USA).

## Micrococcal nuclease digestions and library construction

Micrococcal nuclease digestions were performed with a minimum of two biological replicates as previously described (*Rodriguez et al., 2014*). Briefly, cells were grown to mid-log phase and fixed with 1% formaldehyde. Chromatin was digested with 10, 20, and 40 units of MNase for 10 min. Proper nuclease digestion of DNA was analyzed by agarose gel, and samples with approximately 80% mononucleosomes were selected for library construction. After crosslink reversal, RNase treatment, calf intestine phosphatase (NEB, Ipswich MA, USA) treatment, and proteinase K digestion, mononucleosome-sized fragments were gel-purified and resulting DNA was used to construct libraries with the NuGEN Ovation Ultralow kit per the manufacturer's instructions. Libraries were sequenced at the University of Oregon's Genomics and Cell Characterization Core Facility on an Illumina NextSeq500 on the 37 cycle, paired-end, High Output setting, yielding approximately 10–20 million paired reads per sample.

## Chromatin immunoprecipitation and library construction

Chromatin immunoprecipitation was performed with biological replicates as previously described (*Rodriguez et al., 2014*). Briefly, cells were grown to mid-log phase, fixed with 1% formaldehyde, and lysed by bead-beating in the presence of protease inhibitors. Chromatin was fragmented by shearing in a Bioruptor sonicator (Diagenode, Denville NJ, USA) for a total of 30 min (high output, 3 × 10′ cycles of 30 s on, 30 s off). Sonication conditions were optimized to produce an average fragment size of ~300 base pairs. FLAG-tagged protein was immunoprecipitated using FLAG antibody (Sigma, St. Louis MO, USA) and Protein G magnetic beads (Invitrogen, Waltham MA, USA). After crosslink reversal and proteinase K digestion, DNA was purified using Qiagen MinElute columns and quantified by Qubit High-Sensitivity fluorometric assay. Libraries were prepared using the NuGEN Ovation Ultralow kit by the manufacturer's instructions and sequenced at the University of Oregon's Genomics and Cell Characterization Core Facility on an Illumina NextSeq500 with 37 cycles of paired-end setting, yielding approximately 10 million single-end reads per sample. Only the first read (R1) of each paired read was taken for downstream alignments and processing.

## RNA extraction and library construction

For RNA-Seq (minimum two biological replicates), RNA was purified by hot acid phenol extraction followed by polyA selection and strand-specific library construction using the NuGEN Universal Plus mRNA Kit according to the manufacturer's instructions. Libraries were sequenced on an Illumina NextSeq500 on the 37 cycle, paired-end, high-output setting. Paired-end reads were quality filtered for adapter contamination and low-quality ends using trimmomatic (*Bolger et al., 2014*). After

quality filtering, an average of 10.5 million reads per paired-end sample remained. Surviving reads were mapped to the *S. cerevisiae* reference genome (*Cunningham et al., 2015*) using STAR (V.2.5.3) (*Dobin et al., 2013*). Gene counts were quantified from uniquely aligning reads using HTSeq (V.0.9.1) (*Anders et al., 2015*). Differential gene expression was performed using DESeq2 (V.1.22.2) (*Love et al., 2014*), and expression graphs were generated using ggplot2 (*Wickham, 2016*).

## Data processing and analysis

MNase sequencing data were analyzed as described previously (*McKnight and Tsukiyama, 2015*). Briefly, paired-end reads were aligned to the *S. cerevisiae* reference genome (*Cunningham et al., 2015*) using Bowtie 2 (*Langmead and Salzberg, 2012*) and filtered computationally for unique fragments between 100 and 200 bp. Dyad positions were calculated as the midpoint of paired reads, then dyad coverage was normalized across the *S. cerevisiae* genome for an average read/bp of 1.0. Dyad coverage is displayed in all figures. Nucleosome alignments to transcription Ume6 binding sites were performed by taking average dyad signal at each position relative to all 202 intergenic instances of a Ume6 motif center (*WNGGCGGCWW*). PIC locations were obtained from *Rhee and Pugh, 2012*. For ChIP-Seq data, single-end reads were aligned to the *S. cerevisiae* reference genome with Bowtie 2 and total read coverage was normalized such that the average read at a genomic location was 1.0. ChIP peaks were called using a 400 bp sliding window with a threshold average enrichment within the window of 3.0. Data were visualized using Integrated Genome Browser (*Freese et al., 2016*). The datasets generated during this study are available in the GEO Database with accession code GSE149804.

Sequencing data sets can be accessed in the Gene Expression Omnibus with accession number GSE149804.

## Acknowledgements

The authors thank Christine Cucinotta for helpful comments on the manuscript, and Greg Bowman for helpful discussions about the project. This work was supported by NIH training grants T32 GM007759 (to DAD and OGB) and T32 GM007413 (to DAD and VNT), and by NIGMS grant R01 GM129242 (JNM), the Donald and Delia Baxter Foundation (JNM), and the Medical Research Foundation of Oregon (JNM).

# Additional information

### Funding

| Funder | Grant reference number | Author |
|---|---|---|
| National Institutes of Health | T32 GM007759 | Drake A Donovan<br>Orion GB Banks |
| National Institutes of Health | T32 GM007413 | Drake A Donovan<br>Vi N Truong |
| National Institute of General Medical Sciences | R01 GM129242 | Jeffrey N McKnight |
| Donald E. and Delia B. Baxter Foundation | | Jeffrey N McKnight |
| Medical Research Foundation | | Jeffrey N McKnight |

The funders had no role in study design, data collection and interpretation, or the decision to submit the work for publication.

### Author contributions

Drake A Donovan, Conceptualization, Investigation, Methodology, Writing - original draft, Writing - review and editing; Johnathan G Crandall, Abigail L Vaaler, Investigation, Methodology, Writing - review and editing; Vi N Truong, Thomas B Bailey, Investigation, Methodology; Devin Dinwiddie, Visualization; Orion GB Banks, Formal analysis, Visualization, Writing - review and editing; Laura E

McKnight, Supervision, Investigation, Methodology, Writing - review and editing; Jeffrey N McKnight, Conceptualization, Supervision, Funding acquisition, Investigation, Visualization, Methodology, Writing - original draft, Project administration, Writing - review and editing

### Author ORCIDs
Johnathan G Crandall http://orcid.org/0000-0002-9144-3135
Laura E McKnight https://orcid.org/0000-0002-4322-3066

### Decision letter and Author response
Decision letter https://doi.org/10.7554/eLife.64061.sa1
Author response https://doi.org/10.7554/eLife.64061.sa2

# Additional files

### Supplementary files
- Supplementary file 1. Yeast strains used in this study.
- Supplementary file 2. Plasmids used in this study.
- Transparent reporting form

### Data availability
Sequencing data have been deposited in GEO under accession code GSE149804.

The following dataset was generated:

| Author(s) | Year | Dataset title | Dataset URL | Database and Identifier |
|---|---|---|---|---|
| Donovan DA, Crandall JG, Truong VN, Vaaler AL, Bailey TB, Dinwiddie D, McKnight LE, McKnight JN | 2020 | Basis of specificity for a conserved promiscuous chromatin remodeling protein | https://www.ncbi.nlm.nih.gov/geo/query/acc.cgi?acc=GSE149804 | NCBI Gene Expression Omnibus, GSE149804 |

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
