## [Decision Letter]

**Acceptance summary:**

This is a paradigm shifting study which demonstrates targeting of the Isw2 complex by a bsequence-specific DNA binding protein Ume6. Previously the Isw2 complex was thought to be a promiscuous nucleosome sliding ATPase that would globally space nucleosomes like Chd1 or Isw1. However, the current study demonstrates the Isw2 primarily targets a single nucleosome adjacent to Ume6 binding sites.

**Decision letter after peer review:**

Thank you for submitting your article "Basis of Specificity for a Conserved and Promiscuous Chromatin Remodeling Protein" for consideration by *eLife*. Your article has been reviewed by three peer reviewers, including Jerry L Workman as the Reviewing Editor and Reviewer #1, and the evaluation has been overseen by Kevin Struhl as the Senior Editor. The following individual involved in review of your submission has agreed to reveal their identity: Blaine Bartholomew (Reviewer #3).

The reviewers have discussed the reviews with one another and the Reviewing Editor has drafted this decision to help you prepare a revised submission.

Summary:

Chromatin remodelers use the energy derived from ATP hydrolysis to reposition or evict nucleosomes, thus shaping the chromatin landscape of the cell. In this study, the McKnight lab use creative genetic and genomic approaches to understand how the apparently nonspecific biochemical activity of one such chromatin remodeler, Isw2, is targeted to specific nucleosomes in the budding yeast genome. The use of an isw1/chd1 mutant is a nice approach to remove the effects of spacing factors, and the SpyTag/SpyCatcher approach is a novel idea for artificial recruitment of factors. The bottom line of the study is that small, conserved epitopes in transcription factors act as recruiting elements for Isw2, allowing precise targeting of a nonspecific biochemical activity to specific genomic loci. From a larger perspective, the results lend support to an interacting barrier model of nucleosome positioning, wherein positioning of specific nucleosomes defines the borders of nucleosomal arrays. The data appear to be of high quality and soundly interpreted, and I believe that the results will be of great interest to those interested in chromatin and transcription. There are many questions raised by the results that I believe will drive further investigation into specificity in chromatin remodeling. My one major criticism (not that major in the scheme of things) is that the authors analyze the interesting subsets of their sites, as detailed below. One example is the analysis of the Isw2/Itc2 co-bound sites to the exclusion of the Isw2-alone sites. I think some exploration of these sites would be warranted, as discussed below.

Major comments:

1) In Figure S1C, there is nice correspondence between strong Isw2 K215R binding and Isw2-dependent nucleosome remodeling. However, at PICs where there is no apparent Isw2 remodeling, there does seem to be some Isw2 K215R ChIP-seq signal, albeit at a lower level. Does this potentially represent capture of transient sampling-type interactions, or something else?

2) In Figure 1—figure supplement 3, Ume6 ChIP (WT and DBD alone) is shown at 202 intergenic Ume6 motifs. It is stated that the rows are linked with Figure 1B – it would be nice to see the nucleosome data next to the ChIP data in this panel, as it appears that Ume6 is bound to at some level to the majority of these 202 sites, while Isw2 seems only to be active at the 58 sites of cluster 1. Germane to this point, I of course understand why the authors focused on the cluster 1 sites, but it would be nice to have some speculation on why Isw2 only seems to function at a fraction of Ume6-bound loci. Also, the lengths of the cluster-denoting bars appear to be off here relative to Figure 1B.

3) In Figure 5C, it appears that only a subset of Isw2 sites are bound by Itc1 as well. Again, as with the selection of the 58 Ume6 sites, I understand why the Isw2/Itc1 co-bound sites are selected for further analysis, but the Isw2 sites without Itc1 could be discussed as well. Are these sites non-functional? How does Itc1 ChIP-seq data compare to the Isw2 remodeling activity shown in Figure 1A? How does it compare to Ume6 binding? Does it specify the Isw2-remodeled nucleosomes?

4) Did the authors perform western blots to ensure that their various truncation constructs were stable? This is important for interpretation of the results vs deletions.

5) To summarize the above points, a major thing missing from the discussion is why only subsets of TF binding sites recruit Isw2. For instance, as mentioned above, 58 Ume6 sites seem to specific Isw2 remodeling – what is special about those sites versus the other ~150 sites that appear to be bound by Ume6? It's mentioned briefly in the discussion that only three Swi6 sites were identified as Isw2-recruiting and that this may be tuned by cellular context, but this is quite vague and superficial. More speculation on what differentiates these sites from the TF-bound but non-Isw2 recruiting sites could be included.

---

## [Author Response]

Major comments:1) In Figure S1C, there is nice correspondence between strong Isw2 K215R binding and Isw2-dependent nucleosome remodeling. However, at PICs where there is no apparent Isw2 remodeling, there does seem to be some Isw2 K215R ChIP-seq signal, albeit at a lower level. Does this potentially represent capture of transient sampling-type interactions, or something else?

We believe that this is mostly noise in the ChIP signal, which is exacerbated by the fact that remodelers are notoriously difficult to ChIP. As you have suggested, it is possible that crosslinking in the ChIP experiment captures transient interactions, indicative of a sampling process. We do believe that this sampling likely occurs, as we have observed that Isw2 is able to nonspecifically bind histones and DNA, but that remodeling only occurs when there is a specific/stable interaction. However, we are reluctant to draw specific conclusions from the ChIP data due to the noise in the sample.

2) In Figure 1—figure supplement 3, Ume6 ChIP (WT and DBD alone) is shown at 202 intergenic Ume6 motifs. It is stated that the rows are linked with Figure 1B – it would be nice to see the nucleosome data next to the ChIP data in this panel, as it appears that Ume6 is bound to at some level to the majority of these 202 sites, while Isw2 seems only to be active at the 58 sites of cluster 1. Germane to this point, I of course understand why the authors focused on the cluster 1 sites, but it would be nice to have some speculation on why Isw2 only seems to function at a fraction of Ume6-bound loci. Also, the lengths of the cluster-denoting bars appear to be off here relative to Figure 1B.

We have added the nucleosome position heatmap from Figure 1B to Figure 1—figure supplement 3 so that it is easier to compare to the Ume6 ChIP data. Figure 1B actually shows that both Cluster 1 and Cluster 3 have Isw2-dependent nucleosome shifts; those in Cluster 3 are shifted in the opposite direction, but the Ume6-proximal nucleosomes are consistently more distal to the URS in the absence of Isw2 than in the presence of Isw2. We have updated the language in the second paragraph of the Results section of the text to make this more clear.

We believe that there are several reasons that Isw2 is not active at cluster 2 URS sites. Many of these sites show reduced Ume6 binding, according to our ChIP data, which could in turn lead to the inability to recruit the Isw2 complex. We also suspect that, at least for a subset of these sites, Isw2 is recruited to the motif but the nearest nucleosome is out of reach of the remodeling complex, resulting in an unproductive binding event; we have added this idea to the end of paragraph two of the Results section. We would like to explore this concept further in the future but we feel it is outside the scope of this publication.

We have also adjusted the bar for cluster 1 in Figure 1—figure supplement 3, which was incorrect, to reflect the appropriate clustering as shown in Figure 1B. Thank you for pointing out this discrepancy.

3) In Figure 5C, it appears that only a subset of Isw2 sites are bound by Itc1 as well. Again, as with the selection of the 58 Ume6 sites, I understand why the Isw2/Itc1 co-bound sites are selected for further analysis, but the Isw2 sites without Itc1 could be discussed as well. Are these sites non-functional? How does Itc1 ChIP-seq data compare to the Isw2 remodeling activity shown in Figure 1A? How does it compare to Ume6 binding? Does it specify the Isw2-remodeled nucleosomes?

Due to the difficulty in ChIP-ing Isw2, we aren’t convinced that Isw2 is not actually present at some of the Itc1 sites. We attempted to analyze the Itc1-FLAG ChIP-seq data, but unfortunately we were unable to replicate the exact processing that was used for the original heatmaps, so we could not link the rows to the prior data; therefore it is difficult to draw conclusions from this analysis. However, as we have demonstrated, there are factors other than Ume6 that interact with the Isw2 complex, and we predict that there are many more factors that target Isw2 to specific subsets of sites in the genome.

4) Did the authors perform western blots to ensure that their various truncation constructs were stable? This is important for interpretation of the results vs deletions.

We understand that Western blots are generally standard to show evidence of functional protein, but full-length Ume6 is largely unstructured, making it very unstable outside of the cell so that it degrades rapidly, thus we have been unable to get a satisfactory Western blot of full-length Ume6 or the truncations. However, the ChIP-seq data for the truncated Ume6 constructs recapitulates the data from wild type, which we believe is strong evidence that these truncations are stable, since they occupy the same sites in the genome as full-length Ume6. We have added heatmaps of this data to Figure 1—figure supplement 3 to further strengthen this evidence.

We have purified the Isw2 complex with a more truncated Itc1 construct (∆9-374) for use in some in vitro sliding assays, and the remodeler complex with that truncated Itc1 functions in the same way as the wild type complex; we predict that constructs with fewer amino acids deleted would be less disruptive, and therefore expect all of our truncations are stably folded. We believe the ChIP data for Itc1(1-73) and Itc1(1-132) shown in Figure 5B is also good evidence that these constructs are stable, as they show similar localization to wild type Itc1. There are differences, which we highlight, but it seems unlikely that these differences are the result in protein stability as one would expect a more global change in that case.

5) To summarize the above points, a major thing missing from the discussion is why only subsets of TF binding sites recruit Isw2. For instance, as mentioned above, 58 Ume6 sites seem to specific Isw2 remodeling – what is special about those sites versus the other ~150 sites that appear to be bound by Ume6? It's mentioned briefly in the discussion that only three Swi6 sites were identified as Isw2-recruiting and that this may be tuned by cellular context, but this is quite vague and superficial. More speculation on what differentiates these sites from the TF-bound but non-Isw2 recruiting sites could be included.

You are correct to point out that we did not address why some Ume6 or Swi6 sites don’t exhibit Isw2-remodeling. We have been reluctant to speculate without more concrete data, but we have added language in the final paragraph of the Discussion section of the manuscript to indicate that this is something we have thought about and suggest our ideas on the matter. It is a complex tertiary interaction, which requires assembly of all the necessary elements (the components of the Isw2 complex, along with a recruiting transcription factor, such as Ume6) at the same time. It is likely that there are simply few genomic loci which have all the right pieces of the puzzle in the right place at the right time.